# *Pseudodidymella fagi* in Slovenia: First Report and Expansion of Host Range

**Nikica Ogris** *, **Ana Brglez and Barbara Piškur**

Department of Forest Protection, Slovenian Forestry Institute, 1000 Ljubljana, Slovenia
* Correspondence: nikica.ogris@gozdis.si

**Abstract:** The fungus *Pseudodidymella fagi* is spreading in Europe and causing leaf blotch of European beech, *Fagus sylvatica*. Between 2008 and 2017, outbreaks of *P. fagi* were observed on European beech in Switzerland, Germany (also on *F. orientalis*), Austria, and Slovakia. In Slovenia, leaf blotch symptoms were first observed on *F. sylvatica* in 2018. *P. fagi* was identified as the causal agent of the observed symptoms in Slovenia by morphological examinations together with sequencing of the internal transcribed spacer (ITS) region of the rDNA. This study links the fungus to the expansion of the known distribution of the disease to Slovenia, and based on in vitro pathogenicity trials, also to a new potential host, *Quercus petraea*. The pathogenicity tests confirmed *F. sylvatica* and *F. orientalis* as hosts for *P. fagi*, but not *Castanea sativa*, where pathogenicity to *F. orientalis* was proved for first time in vitro. Although Koch's postulates could not be proven for *C. sativa*, it seems to be partially susceptible in vitro because some of the inoculation points developed lesions. Additionally, damage to *Carpinus betulus* related to *P. fagi* near heavily infected beech trees was observed in vivo but was not tested in laboratory trials. Based on the results and our observations in the field, it is likely that *P. fagi* has a wider host range than previously thought and that we might be witnessing host switching.

**Keywords:** *Pycnopleiospora fagi*; leaf blotch; pathogenicity test; inoculation test; *Fagus orientalis*; *Fagus sylvatica*; *Quercus petraea*; mycopappus-like propagule; *Carpinus betulus*; host switching

---

## 1. Introduction

*Pseudodidymella fagi* C.Z. Wei, Y. Harada & Katum. is an ascomycete that was described in Japan in 1993, where it was associated with leaf blotch of Japanese beech, *Fagus crenata* Blume [1]. The sexual morph is formed over winter, and ascospores mature in late spring and initiate new infections of young beech leaves. Its pycnopleiospora asexual morph is formed under favourable conditions on necrotic leaf spots a few weeks after the initial infection has occurred. Conidia are responsible for secondary infections in summer [1].

The first observation of the pathogen in Europe was in Switzerland in 2008, where it was identified on European beech (*Fagus sylvatica* L.) as a new host [2]. In 2016, *P. fagi* was also identified in southern Germany on *F. sylvatica* and on Oriental beech (*Fagus orientalis* Lipsky) in the Munich Botanical Garden [2] and in Austria [3]. In 2017, the disease was recorded in Slovakia [4]. Reports of *P. fagi* spread in Europe indicate that it possesses the fundamental characteristics of an invasive alien species. However, this hypothesis still needs further testing [2].

In 2018, leaf blotch of European beech was observed in central Slovenia. Leaves of *F. sylvatica* showed dark brown, irregularly shaped or confluent necrotic spots. Necrotic spots were more numerous on young plants and the lower branches of mature trees. The intensity of leaf blotch decreased progressively with increasing distance from the ground. To identify the causal agent, morphological and molecular analyses together with pathogenicity trials on *F. sylvatica* were used to identify and confirm the pathogen. In addition, pathogenicity tests were performed on other

representative species of the Fagaceae family, i.e., *F. orientalis*, *Castanea sativa* Mill., and *Quercus petraea* (Matt.) Liebl., to test the possibility of a wider host range of *P. fagi*.

## 2. Materials and Methods

### 2.1. Sample Collection and Distribution of Pseudodidymella fagi

Leaves of *F. sylvatica* infected by *P. fagi* were found and collected in August 2018 in Kamniška Bela, in central Slovenia (46.32020° N, 14.60161° E, 620 m a.s.l.). The infected leaves were collected and deposited in the Mycotheca and Herbarium of the Slovenian Forestry Institute (LJF 7014–LJF 7016). In addition, further collection of European beech leaves from the leaf litter was done in spring 2019 to identify the sexual morph (LJF 7017).

Visually healthy leaves from representatives of the Fagaceae family were collected on 23 August 2018, for pathogenicity trials (Table 1).

**Table 1.** Locations of samples collected for pathogenicity trials.

| Plant Species | Location | Latitude (° N) | Longitude (° E) | Elevation (m a.s.l.) |
|---|---|---|---|---|
| *Fagus sylvatica* | Ljubljana | 46.05195 | 14.47859 | 312 |
| *Fagus orientalis* | Volčji Potok | 46.18361 | 14.61061 | 384 |
| *Quercus petraea* | Ljubljana | 46.05277 | 46.05277 | 322 |
| *Castanea sativa* | Ljubljana | 46.05257 | 14.47903 | 319 |

The Slovenia Forest Service (SFS) was used to report symptoms and send samples to confirm the pathogen and to determine its distribution in Slovenia.

### 2.2. Morphological Analyses

Symptomatic leaves were examined, photo-documented, and analyzed with an Olympus SZX16 stereomicroscope (Tokyo, Japan) with an Olympus UC90 camera (Münster, Germany), an Olympus BX53 microscope (Tokyo, Japan) with an Olympus DP26 camera (Tokyo, Japan), and an Olympus CellSens Standard v1.18 software.

Isolations from single or multiple conidium and/or ascospore were carried out by aseptically transferring single propagules from leaves to fresh malt-extract agar plates (MEA; 2% malt extract, 1.5% agar; Becton Dickinson, Sparks, MD, USA) and to potato dextrose agar plates (PDA; 3.9% potato dextrose agar; Difco[TM], Becton Dickinson, Sparks, MD, USA) with the tip of a scalpel and incubating them at 24 °C in the dark. In total, ten isolates from fruiting bodies were acquired. Representative isolates were deposited in the culture collection of the Laboratory of Forest Protection (Slovenian Forestry Institute, Ljubljana, Slovenia, ZLVG683 and ZLVG684).

For isolations from lesions on leaves, a procedure similar to that used by Gross, Beenken, Dubach, Queloz, Tanaka, Hashimoto and Holdenrieder [2] was used—leaves were surface sterilized for 30 s in 80% ethanol, and squares of approximately 5 mm$^2$ of symptomatic tissue per leaf were excised and transferred to MEA plates amended with 100 mg L$^{-1}$ streptomycin (Sigma-Aldrich, St. Louis, MO, USA).

### 2.3. DNA Extraction, DNA Sequencing, and Identification

Genetic analyses were used to confirm the identity of the fungal propagules and obtained isolates. DNA was extracted from the mycelium, scraped from the MEA agar plates, or was extracted from 3–6 conidiomata carefully detached directly from infected leaves. Genomic DNA was extracted with a commercial NucleoSpin® Plant II kit (Macherey Nagel, Düren, Germany) following the manufacturer's protocol, after homogenizing the fungal material with a lysing matrix A tube (MP Biomedicals, Solon, OH, USA) using a Precellys Evolution device (Bertin Technologies, Montigny-le-Bretonneux, France). The internal transcribed spacer (ITS) region of the ribosomal DNA (rDNA) was amplified with primer

pair ITS1 and ITS4 [5]. The final volume of the amplification mixtures was 50 μL and contained 25 μL AmpliTaq Gold® 360 Master Mix (Applied Biosystems, Foster City, CA, USA), 1 μL 360 GC Enhancer (Applied Biosystems), 3 μL each of 10 μM primers (Sigma-Aldrich), 4 μL of approximately 10 μg/mL extracted DNA, and 14 μL of nuclease-free water. The PCR conditions were as follows: 95 °C for an initial 10-min denaturation, 35 cycles of 94 °C for 1 min, annealing at 61.5 °C for 1 min and extension at 72 °C for 1 min, and a final extension at 72 °C for 10 min. The obtained PCR products were cleaned using a Wizard SV Gel and PCR Clean-Up System (Promega, Madison, WI, USA) and sequenced at a sequencing facility (Eurofins, Ebersberg, Germany) in both forward and reverse directions using the same primers as for the PCRs. Sequences were visualised and manually edited using Geneious Prime® v.2019.0.3. (Biomatters Ltd., Auckland, New Zealand). Each sequence was used to perform individual nucleotide–nucleotide searches with the megablast algorithm from the NCBI website on 12 July 2019 [6]. Representative sequences were deposited in the GenBank database.

### 2.4. Pathogenicity Tests

Pathogenicity tests were performed on representative tree species from the family Fagaceae: *Fagus sylvatica*, *Fagus orientalis*, *Castanea sativa*, and *Quercus petraea*. Visually healthy leaves of *F. sylvatica*, *C. sativa*, and *Q. petraea* were collected in mixed broadleaved forests on 23 August 2018 (Table 1). The leaves were rinsed with autoclaved distilled water and placed on moistened paper tissues in sterile glass plates (19 cm diameter, 4 cm height) as follows: four beech leaves per plate, four oak leaves per plate, and one sweet chestnut leaf per plate for a total of eight leaves per host. For inoculation, firstly three to six *P. fagi* propagules on symptomatic *F. sylvatica* leaves were morphologically and molecularly identified, and secondly the remaining propagules from infected *F. sylvatica* leaves were carefully transferred with a sterile scalpel to the upper side of the leaves, which were included in the pathogenicity trials. Leaves of *F. sylvatica*, *F. orientalis*, and *Q. petraea* were inoculated with five propagules and *C. sativa* leaves with ten propagules. In total, there were 40 inoculation points for *F. sylvatica*, *F. orientalis*, and *Q. petraea*, and 80 inoculation points for *C. sativa*. Four non-inoculated healthy leaves of each species represented controls. After inoculation, leaves were incubated in petri dishes under daylight conditions at room temperature (22–25 °C). Paper tissues were periodically remoistened with sterile distilled water to avoid desiccation. The leaves were regularly inspected using an Olympus SZX16 stereomicroscope to monitor the infection process. After 22 days of incubation, re-isolations were carried out from the freshly developing conidiomata and from necrotic leaf tissues developing in the immediate vicinity of the inoculation points. Additional isolations from non-symptomatic leaves and controls were carried out to verify that no latent *P. fagi* infections were present. For isolations from freshly developing conidiomata and for re-isolations from leaves, the same procedures were used as described in morphological analyses in Section 2.2. The following number of samples were used for re-isolations per host: 36 samples for *F. sylvatica*, 31 samples for *F. orientalis*, 21 samples for *C. sativa*, and 18 samples for *Q. petraea*. Plates were incubated at 24 °C in the dark and regularly checked for mycelium growth for two weeks (negative controls for 29 days). Any outgrown mycelium was immediately transferred to fresh PDA and MEA plates. The isolates were grouped into morphotypes based on colony characteristics, and representative isolates from each morphotype were identified based on the ITS rDNA sequence.

## 3. Results

### 3.1. Distribution and Identification of Pseudodidymella fagi

The first symptoms of *P. fagi* on European beech in Slovenia were observed in August 2018 in Kamniška Bela (46.32020° N, 14.60161° E) and in Ljubljana (46.05195° N, 14.47859° E). Both sites were located near streams in mixed managed forests and were characterized by high humidity. With the help of foresters from the Slovenia Forest Service, the disease was confirmed to be present in a large part of Slovenia (Figure 1).

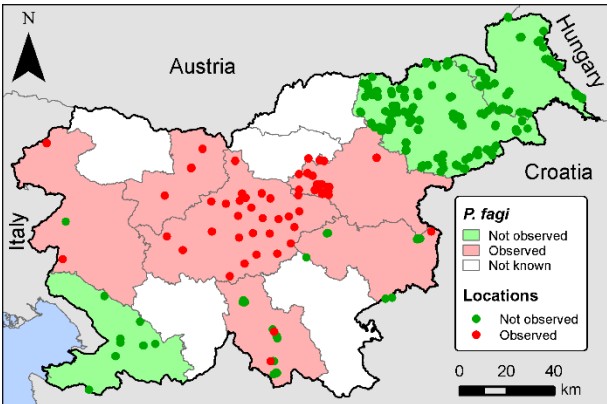

**Figure 1.** Distribution map of *Pseudodidymella fagi* in Slovenia by regional units of the Slovenia Forest Service along with locations that were checked for presence of the pathogen.

Leaves of *F. sylvatica* showed dark brown necrotic spots (Figure 2a). Necrotic spots were irregularly shaped or confluent and usually had a sharp and darker border with healthy tissues (Figure 2b) and with mycopappus-like propagules (asexual morph) on the surface (Figure 2c). The propagules were globose, multicellular, hyaline, or very pale yellow, measuring 142–239 (ø = 193) µm in diameter (without appendages; Figure 2d). They bore numerous (>130) hyaline, unbranched, up to 4-septate, 84–165 (ø = 129)-µm long, and 4.3–6.4 (ø = 5.2)-µm wide appendages (Figure 2e) originating from globular, monilioid, hyaline, or pale yellow hyphae, which measured 8.6–16.1 (ø = 11.6) µm in diameter (Figure 2f,g).

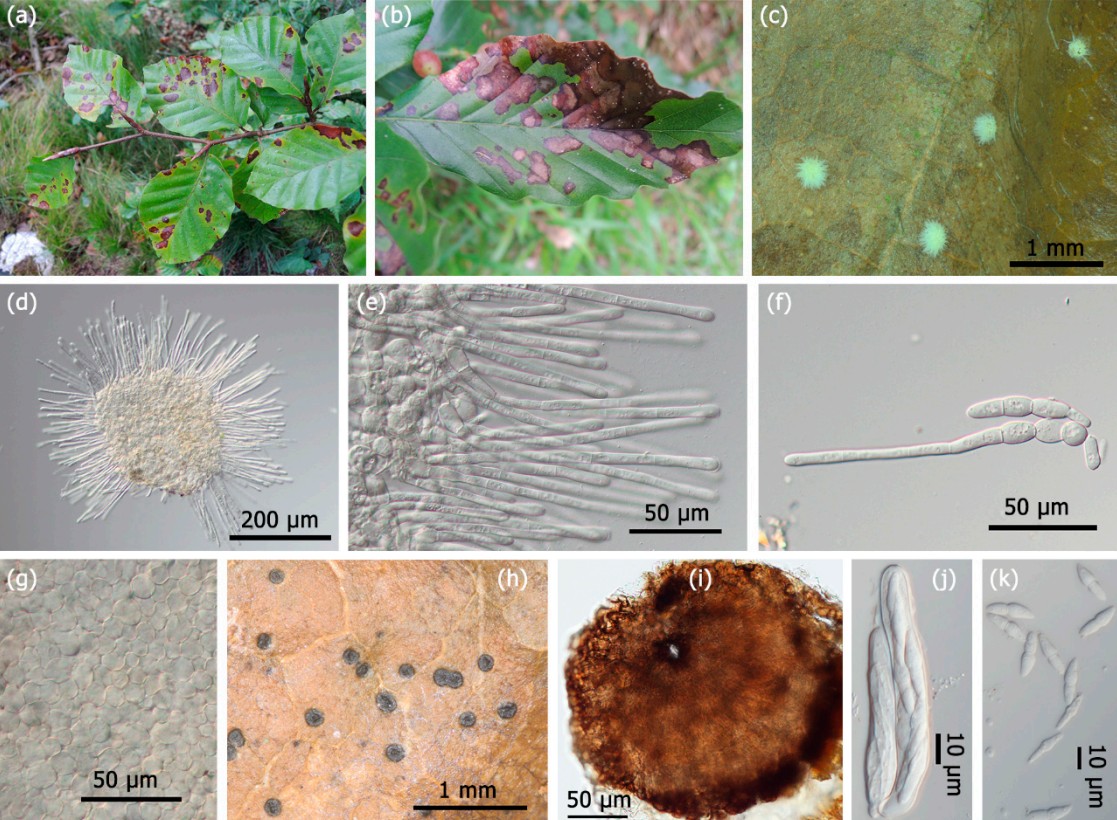

**Figure 2.** Symptoms and morphology of *Pseudodidymella fagi*: (**a**,**b**) Necrotic leaf spots; (**c**) necrotic leaf spot covered with mycopappus-like propagules; (**d**) macroscopic view of mycopappus-like propagule, conidium; (**e**) septate appendages; (**f**) basal cells of an appendage; (**g**) globular cells in the central part of a propagule; (**h**) ascomata on upper leaf surface; (**i**) ostiole of an ascoma; (**j**) asci; (**k**) ascospores.

On 19 April 2019, at the time of European beech leaf flush, the sexual morph of *P. fagi* was observed in the litter on *F. sylvatica* leaves from the previous year. Ascomata were dark brown to black (Figure 2h,i), subglobose to lenticular, semi-immersed in the host tissue, measuring 121–240 (ø = 178) μm in diameter with a sunken centre in dry state. Asci were 51–83 (ø = 63) μm long and 9.1–11.6 (ø = 10.2) μm wide, bitunicate, clavate to cylindrical, with round apex, short stalked (Figure 2j). Ascospores were 17.4–25.5 (ø = 21.6) μm long and 4.0–6.5 (ø = 5.5) μm wide, hyaline, mostly 1-septate, constricted at the septum, distinctly pointed but rounded at the apex (Figure 2k). Morphological identification corresponded to the description of *Pseudodidymella fagi* [1,2,4,7].

The *P. fagi* cultures on PDA were pulvinate, olivaceous-grey with black exudates on the upper side and dark olivaceous to blackish on the reverse. The *P. fagi* colonies on MEA were mostly flat, blackish to dark olivaceous, patchily floccose and greyish. The growth rate of *P. fagi* colonies measured 33–54 (ø = 42) mm in 21 days at 24 °C in the dark on PDA.

The morphological identification of both fungal propagules and isolated cultures was confirmed based on ITS rDNA sequence comparison. The sequences from propagules (MN170774) and from mycelial isolates ZLVG683 (MN170775) and ZLVG684 (MN170776) showed 100% identity (448 bp/448 bp) with the sequence obtained from *P. fagi* holotype H2579 (GenBank Acc. No. LC150787).

### 3.2. Pathogenicity Tests

#### 3.2.1. Fagus sylvatica

No necrotic lesions developed on negative controls of *F. sylvatica* (Figure 3a). On inoculated leaves of *F. sylvatica*, necrotic lesions started to develop after two days underneath or in the immediate vicinity of the inoculated propagule (Figure 3c), and microscopic examination showed numerous appressoria on the leaf surface (Figure 3e). After 21 days, 70% of inoculations had a typical, light to dark brown necrosis under the propagule (Figure 3b). Fourteen days after inoculation, numerous mycopappus-like propagules were observed on 10 out of the 30 lesions observed (Figure 3d). From 12 isolations of mycopappus-like propagules, five isolates similar to *P. fagi* were obtained. Re-isolations from 21 lesions of *F. sylvatica* yielded seven isolates (out of 34) morphologically resembling *P. fagi* (Figure 3g–i) and 27 other isolates grouped into seven morphotypes. The ITS sequence of the representative isolate of the *P. fagi* morphotype (MN170779) obtained from symptomatic *F. sylvatica* leaves was 100% identical with *P. fagi* sequence LC150787. The most frequently isolated fungus from necrosis on inoculated *F. sylvatica* leaves was *Colletotrichum* sp., which was obtained from 13 isolations (out of 34).

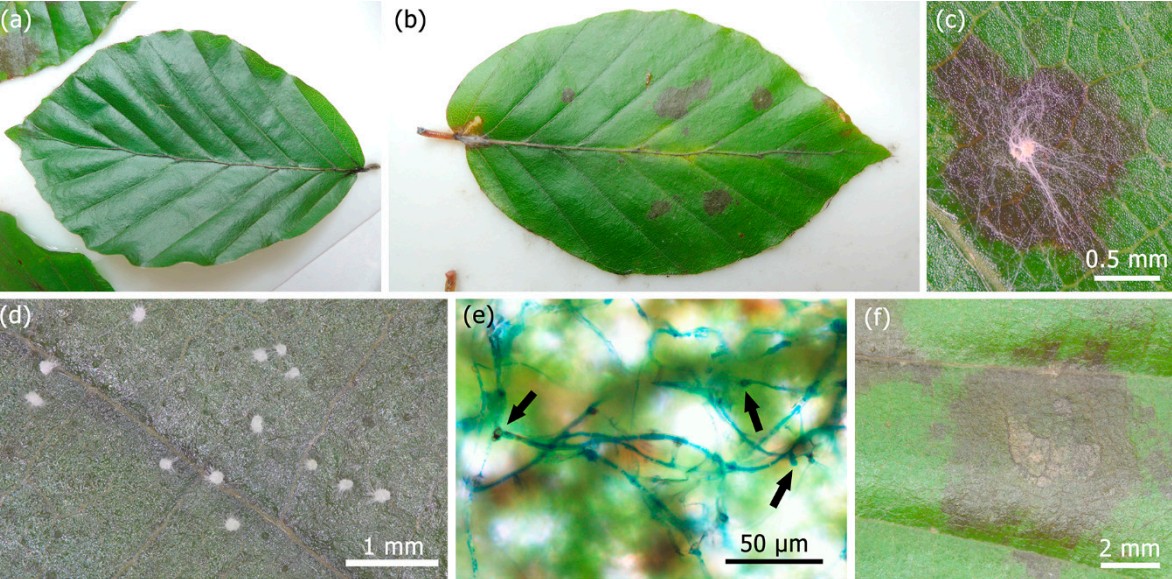

**Figure 3.** *Cont.*

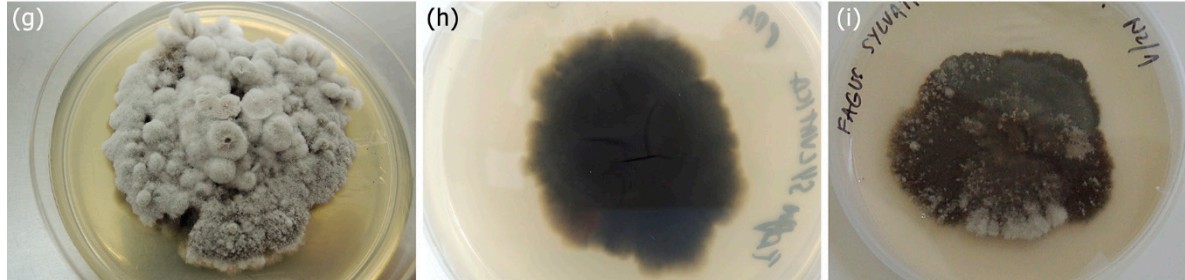

**Figure 3.** Pathogenicity test on *Fagus sylvatica*: (**a**) Control on the 21st day; (**b**) necrotic leaf spots on the 21st day; (**c**) a lesion developed under a propagule on the 2nd day; (**d**) fresh mycopappus-like propagules developed on a necrotic spot on the 14th day; (**e**) appressorium formation (arrows) on the 11th day; (**f**) disintegration of the inoculated propagule on the 21st day; (**g,h**) upper and reverse side of culture re-isolated from newly formed mycopappus-like propagule on PDA, respectively; (**i**) re-isolation from newly formed mycopappus-like propagule on MEA.

### 3.2.2. *Fagus orientalis*

No necrotic lesions developed on negative controls of *F. orientalis* (Figure 4a). On inoculated leaves of *F. orientalis*, lesions started to develop after two days (Figure 4c). Altogether, 74% of inoculations developed necrosis directly underneath or near the inoculated propagules. The size of the lesions gradually increased (Figure 4b,d). Newly formed mycopappus-like propagules started to develop from three lesions after 14 days (Figure 4e). Five pure cultures were obtained from isolations of freshly formed mycopappus-like propagules, similar to *P. fagi*. Re-isolations from symptomatic tissues of *F. orientalis* yielded two pure cultures (out of 30) morphologically resembling *P. fagi* (Figure 4f,g). The ITS sequence of the representative isolate of the *P. fagi* morphotype (MN170777) obtained from symptomatic *F. orientalis* leaves was 100% identical to *P. fagi* sequence LC150787. The most frequently isolated fungus from *F. orientalis* leaves was *Colletotrichum* sp. (18 out of 30 isolates).

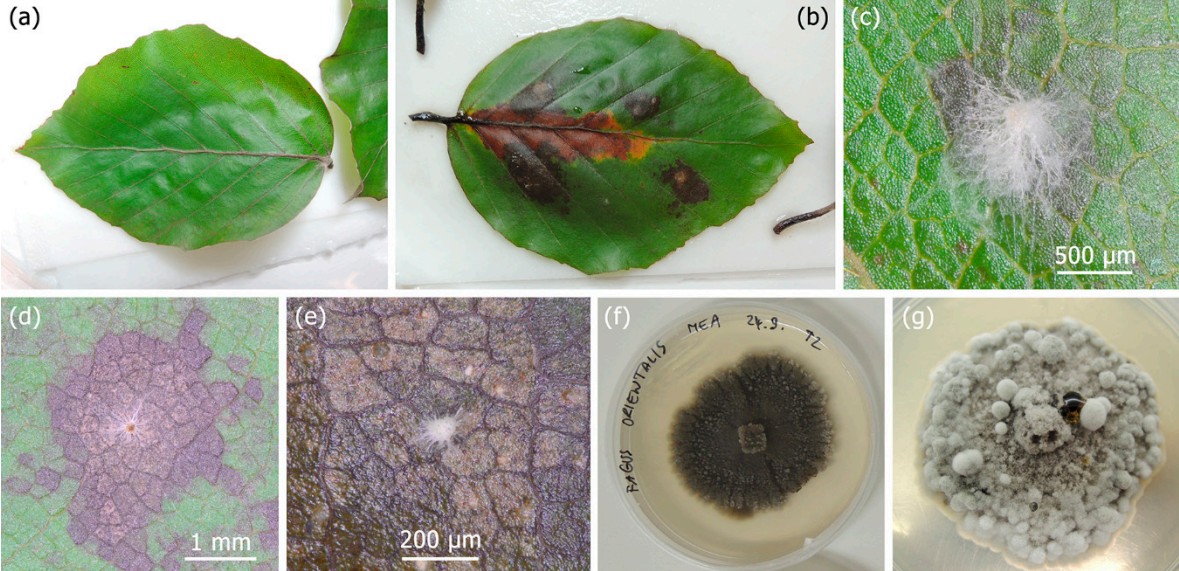

**Figure 4.** Pathogenicity test on *Fagus orientalis*: (**a**) Control on the 21st day; (**b**) necrotic leaf spots on the 21st day; (**c**) a lesion developed under a propagule on the 2nd day; (**d**) lesion under a propagule on the 11th day; (**e**) newly formed mycopappus-like propagule on a necrotic spot on the 14th day; (**f,g**) re-isolation from newly formed mycopappus-like propagule on MEA and PDA, respectively.

### 3.2.3. *Quercus petraea*

No necrotic lesions developed on negative controls of *Q. petraea* (Figure 5a). Lesions developed on 28% of inoculation points, i.e., directly underneath or near the inoculated propagules (Figure 5b,c). No mycopappus-like propagules developed on the lesions during the experiment. Re-isolations from symptomatic tissues of *Q. petraea* yielded one pure culture (out of 10) morphologically resembling *P. fagi* (Figure 5d,e) and nine isolates, which were grouped into seven morphotypes. The ITS sequence of the isolate of the *P. fagi* morphotype (MN170778) obtained from symptomatic *Q. petraea* leaves was 100% identical to *P. fagi* sequence LC150787. The most frequently isolated fungus from *Q. petraea* inoculated leaves was *Paraphaeosphaeria* sp. (two out of ten isolates).

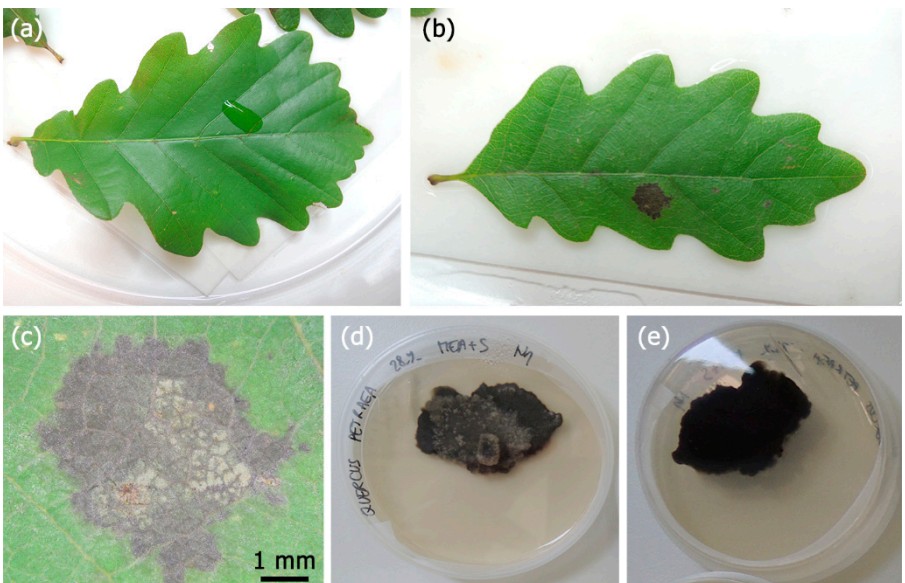

**Figure 5.** Pathogenicity test on *Quercus petraea*: (**a**) Control on the 21st day; (**b**) necrotic leaf spot on the 21st day; (**c**) disintegration of the inoculated propagule on the 21st day; (**d**,**e**) upper and reverse side of culture re-isolated from necrotic leaf spot on MEA, respectively.

### 3.2.4. *Castanea sativa*

No necrotic lesions developed on negative controls of *C. sativa* (Figure 6a). Only 19% of inoculation points developed lesions (Figure 6d) and most of the inoculated leaves stayed healthy (Figure 6b,c). No mycopappus-like propagules developed on the observed lesions on artificially inoculated *C. sativa* leaves. Re-isolations from developed lesions on *C. sativa* leaves yielded no cultures that would morphologically resemble *P. fagi*, however yielded nine isolates, which were classified into three morphotypes. The most frequently isolated fungus from *C. sativa* leaves was *Gnomoniopsis smithogilvyi* L.A. Shuttlew., E.C.Y. Liew & D.I. Guest.

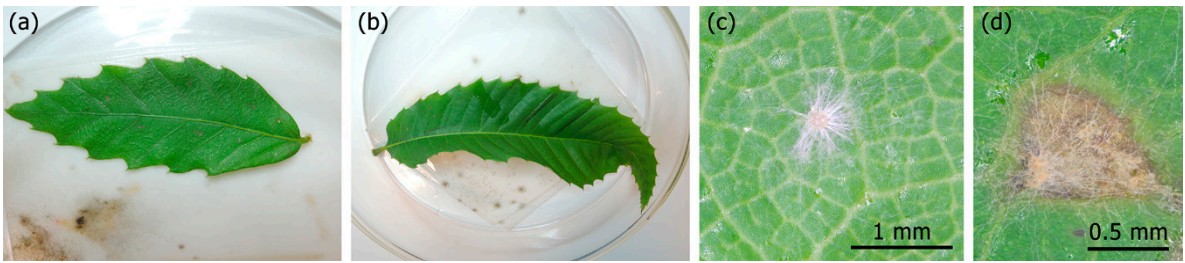

**Figure 6.** Pathogenicity test on *Castanea sativa*: (**a**) Control on the 21st day; (**b**) inoculated leaf on the 21st day; (**c**) germination of a propagule on the 2nd day; (**d**) necrotic leaf spot on the 21st day.

## 4. Discussion

The disease symptoms caused by *Pseudodidymella fagi* on European beech were observed across a large part of Slovenia, which is now the fifth European country with a known presence of *P. fagi*. Susceptible hosts for *P. fagi* are known to be *F. crenata*, *F. sylvatica*, and *F. orientalis*, but the pathogenicity tests conducted in this study proved that *P. fagi* is also a potential pathogen for *Q. petraea*. Therefore, this study is the first confirmation of Koch's postulates for *P. fagi* towards *F. orientalis* and *Q. petraea*.

It is still not clear whether *P. fagi* is an overlooked pathogen that is causing more evident damage throughout Europe due to climatic changes or a newly introduced invasive species [2]. It is probable that the similar symptomatology compared to *Apiognomonia errabunda* (Roberge ex Desm.) Höhn. concealed the earlier discovery of *P. fagi* in Slovenia. Numerous recent reports of *P. fagi* occurring in Europe support the hypothesis that *P. fagi* was introduced into Europe, most likely from Asia [1,7], but further genetic studies are needed to appropriately address this question.

Visually healthy leaves of the four tree species were inoculated with propagules of *P. fagi*. Even though 19–74% of the inoculation spots developed lesions under or in the immediate vicinity of the propagule, re-isolations from those regions did not always yield the culture of *P. fagi*. Furthermore, re-isolation of *P. fagi* after artificial infection proved to be difficult and resulted in a very low success rate for all tested species. On the other hand, numerous other fungi were re-isolated from the necrotic leaf spots under the inoculation points, such as *Colletotrichum* sp. from European beech and Oriental beech, *Paraphaeosphaeria* sp. from sessile oak, and *Gnomoniopsis smithogilvyi* from sweet chestnut. Some of the identified species are well-known endophytes [8–11]. We speculate that endophytes could have an important role in competing with *P. fagi* for the same substrate or could have a role in the development of necrosis. Additionally, endophytes could play a decisive role in combating the future spread of *P. fagi* across Europe, as is also known for other endophytic fungi [12,13]. As such, the role of endophytes in the development and spread of the disease caused by *P. fagi* should be investigated in the future.

Our study reports the first finding of *P. fagi* in Slovenia and its further spread in Europe towards the south. We confirmed the pathogenicity of *P. fagi* towards *F. sylvatica*. This is the first study that confirmed *F. orientalis* as a host, in agreement with Koch's postulates. Our in vitro inoculation experiment showed potential pathogenicity towards *Q. patraea* and *C. sativa*, which is the first report of new potential hosts for *P. fagi* in Europe. However, no characteristic symptoms for *P. fagi* were observed in the field on *Q. petraea* and *C. sativa* leaves. Re-isolations of the pathogen from inoculated *C. sativa* leaves were not successful. However, *C. sativa* seems to be partially susceptible in vitro because 19% of inoculation points developed lesions. Koch's postulates could not be fulfilled for *C. sativa*; nevertheless, it is possible that the observed necroses were caused by *P. fagi* and its re-isolation failed because re-isolations of *P. fagi* proved to be difficult for all tested species. In the field, numerous heavily damaged leaves of common hornbeam (*Carpinus betulus* L.) were observed in the immediate vicinity of European beech heavily infected by *P. fagi*. Macroscopic examination showed that some necrotic lesions on *C. betulus* leaves developed underneath or in the immediate vicinity of the mycopappus-like propagules of *P. fagi*. However, no newly developed propagules were found on those lesions. We hypothesize that *C. betulus* could be another possible host, where *P. fagi* acts as a weak pathogen. Therefore, extended pathogenicity tests should be performed that would include representatives from the Fagales order, i.e., from the following genera: *Alnus*, *Betula*, *Carpinus*, *Corylus*, and *Ostrya*. We are most likely witnessing host jumps and host switching of *P. fagi* [2,7,14], and close monitoring of this arising pathogen throughout Europe would clarify the situation of the disease on other hosts.

## 5. Conclusions

*P. fagi* is reported for first time in Slovenia. The pathogenicity of *P. fagi* towards *F. sylvatica* was confirmed once again. Pathogenicity towards *F. orientalis* and *Q. petraea* was confirmed for first time in vitro. *C. sativa* seems to be partially susceptible in vitro. Based on the results and our observations

in the field, it is likely that *P. fagi* has a wider host range than previously thought and that we are most likely witnessing host jumps and host switching of *P. fagi*.

**Author Contributions:** Conceptualization, N.O.; data curation, N.O., A.B., and B.P.; formal analysis, N.O. and B.P.; funding acquisition, N.O.; investigation, N.O., A.B., and B.P.; methodology, N.O. and B.P.; project administration, N.O.; resources, N.O., A.B., and B.P.; supervision, N.O.; Validation, N.O., A.B., and B.P.; visualization, N.O.; writing—original draft, N.O.; writing—review and editing, N.O., A.B., and B.P.

**Funding:** This research was funded by the Ministry of Agriculture, Forestry, and Food (Public Forestry Service) and the Slovenian Research Agency (Research Program P4-0107).

**Acknowledgments:** We are grateful to Špela Jagodic and Zina Devetak for technical assistance and to foresters from the Slovenia Forest Service who helped with providing samples and information about the distribution range of the disease in Slovenia. We thank Thomas Cech at the Austrian Research Centre for Forest in Vienna for providing initial help and encouragement for searching for the disease. We are grateful to Jan Nagel for providing language help. We are especially grateful to two reviewers for their constructive comments, which improved the manuscript greatly.

**Conflicts of Interest:** The authors declare no conflict of interest. The funders had no role in the design of the study; in the collection, analyses or interpretation of the data; in the writing of the manuscript; or in the decision to publish the results.

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
