# Peer review of "Pseudodidymella fagi in Slovenia: First Report and Expansion of Host Range"

_forests, doi:10.3390/f10090718_

Round 1

Reviewer 1 Report

Comments to the authors
Short title: Pseudodidymella fagi in Slovenia (to Forests)
Authors: Ogris et al.

This is an important paper to know the host range and distribution in Europe of Pseudodidymella fagi, the leaf blotch fungus of European beech.
This manuscript can be accepted for publication after several minor revisions, as indicated below.

1) row 27: Please use sexual morph rather than teleomorph (Similarly, use asexual morph rather than anamorph).

2) row 28: Pycnopleiospora fagi should be treated as a synonym of Pseudodidymella fagi.
Therefore, "Its anamorph, Pycnopleiospora fagi C. Z. Wei, Y Harada & Katum., is formed..." should be changed to;
"Its pycnopleiospora [do not use italics] asexual morph is formed..."

3) row 51: sexual morph (not teleomorph).

4) row 159: "The sequences from conidiomata (MN170774) and from isolates ZLVG683 (MN170775) and ZLVG684..." should be changed to
"The sequences from conidium [or propagule] (MN170774) and from [mycelial] isolates ZLVG683 (MN170775) and ZLVG684...
Please clarify the sources of isolates.

5) row 173: The most frequently isolated fungus from necrosis on inoculated F. sylvatica leaves was Colletotrichum sp., which was obtained from 13 isolations (out of 34) (Table S1).
row 192: The most frequently isolated fungus from F. orientalis leaves was Colletotrichum sp. (18 out of 30 isolates) (Table S1).
row 207: The most frequently isolated fungus from Q. petraea inoculated leaves was Paraphaeosphaeria sp. (two out of five isolates) (Table S1).
row 217: Re-isolations from developed lesions on C. sativa leaves yielded no cultures that would morphologically resemble P. fagi and nine isolates, which were classified into three morphotypes. The most frequently isolated fungus from C. sativa leaves was Gnomoniopsis smithogilyvi (Table S1).
row 235-244: Visually healthy leaves of F. sylvatica were inoculated with propagules of P. fagi. Even though...As such, the role of endophytes in the development and spread of the disease caused by P. fagi should be investigated in the future.
In my personal opinion, these sentences, as well as Table S1, do not provide useful information to reader of this paper, and thus, should be deleted from the manuscript.
It is not uncommon to obtain non-pathogenic fungi from a particular lesion of a diseased plant.
Depending on the condition of the diseased plant, as well as sterilization time of leaf surface, various unrelated fungi including saprophytes will be isolated.
If the authors treat them as endophytic fungi and discuss their competition with the pathogen, additional experiments will be needed to prove it.
Therefore, I think that discussion about role of endophyte in this manuscript (row 235-244) is questionable and appear over-interpreted.

6) row 252: Re-isolations of the pathogen from artificially infected C. sativa leaves were not successful.
I think this result is natural because the "pathogen (but not pathogen)" did not infect this plant (thus, should not be treated as "infected leaves").

7) row 259: Betula, Alnus, Carpinus, Ostrya, and Corylus
Please arrange alphabetically.

Author Response

Point-by-point replies to the comments of Reviewer #1

(Legend: Bold = reviewer comment, “>>” = authors reply)

This is an important paper to know the host range and distribution in Europe of Pseudodidymella fagi, the leaf blotch fungus of European beech.
This manuscript can be accepted for publication after several minor revisions, as indicated below.

1) row 27: Please use sexual morph rather than teleomorph (Similarly, use asexual morph rather than anamorph).

>> We changed “teleomorph” to “sexual morph” (lines 28, 57, 155) and “anamorph” to “asexual morph” (lines 30, 141) throughout the manuscript.

2) row 28: Pycnopleiospora fagi should be treated as a synonym of Pseudodidymella fagi.
Therefore, "Its anamorph, Pycnopleiospora fagi C. Z. Wei, Y Harada & Katum., is formed..." should be changed to;
"Its pycnopleiospora [do not use italics] asexual morph is formed..."

>> We agree. This is synonym. We changed it as suggested: “Its pycnopleiospora asexual morph is formed...”

3) row 51: sexual morph (not teleomorph).

>> Changed as suggested

4) row 159: "The sequences from conidiomata (MN170774) and from isolates ZLVG683 (MN170775) and ZLVG684..." should be changed to
"The sequences from conidium [or propagule] (MN170774) and from [mycelial] isolates ZLVG683 (MN170775) and ZLVG684...
Please clarify the sources of isolates.

>> We changed the sentence as suggested. Additionally, plural of propagule was used: “The sequences from propagules (MN170774) and from mycelial isolates ZLVG683 (MN170775) and ZLVG684 (MN170776)…”

5) row 173: The most frequently isolated fungus from necrosis on inoculated F. sylvatica leaves was Colletotrichum sp., which was obtained from 13 isolations (out of 34) (Table S1).
row 192: The most frequently isolated fungus from F. orientalis leaves was Colletotrichum sp. (18 out of 30 isolates) (Table S1).
row 207: The most frequently isolated fungus from Q. petraea inoculated leaves was Paraphaeosphaeria sp. (two out of five isolates) (Table S1).
row 217: Re-isolations from developed lesions on C. sativa leaves yielded no cultures that would morphologically resemble P. fagi and nine isolates, which were classified into three morphotypes. The most frequently isolated fungus from C. sativa leaves was Gnomoniopsis smithogilyvi (Table S1).
row 235-244: Visually healthy leaves of F. sylvatica were inoculated with propagules of P. fagi. Even though...As such, the role of endophytes in the development and spread of the disease caused by P. fagi should be investigated in the future.
In my personal opinion, these sentences, as well as Table S1, do not provide useful information to reader of this paper, and thus, should be deleted from the manuscript.

>> Table S1 was deleted from the text. However, we kept the data about most frequently re-isolated fungi because they are actual observations during our experiments and it gives added value to the manuscript.

It is not uncommon to obtain non-pathogenic fungi from a particular lesion of a diseased plant.
Depending on the condition of the diseased plant, as well as sterilization time of leaf surface, various unrelated fungi including saprophytes will be isolated.
If the authors treat them as endophytic fungi and discuss their competition with the pathogen, additional experiments will be needed to prove it.
Therefore, I think that discussion about role of endophyte in this manuscript (row 235-244) is questionable and appear over-interpreted.

>> We rephrased this part in the discussion (lines 253 to 265) where we emphasize that the hypothesis about the role of endophytes is solely a speculation, and we have already stated in the first version of the manuscript that “the role of endophytes in the development and spread of the disease caused by P. fagi should be investigated in the future” (lines 263–265).

6) row 252: Re-isolations of the pathogen from artificially infected C. sativa leaves were not successful.
I think this result is natural because the "pathogen (but not pathogen)" did not infect this plant (thus, should not be treated as "infected leaves").

>> We agree, and this part was changed from “artificially infected C. sativa leaves” to “inoculated C. sativa leaves” (new line 288).

7) row 259: Betula, Alnus, Carpinus, Ostrya, and Corylus
Please arrange alphabetically.

>> We arranged the names alphabetically to: “Alnus, Betula, Carpinus, Corylus, and Ostrya

Reviewer 2 Report

Ogris et al. report the recently detected pathogen P. fagi from Slovenia for the first time. Besides this new first report, authors also performed pathogenicity tests on three different potential host trees of P. fagi and they report, that one of these – Quercus petraea -  is partially susceptible in vitro. In addition, they also report on natural infections of Carpinus betulus near heavily infested Fagus sylvatica trees.

They paper is written and structured quite clear and sound. The English is mostly fine with some minor grammar mistakes here and there. Experiments were carried out according to pathological standards, i.e. they conducted negative controls, made surface sterilizations before isolations and reisolations, identified the isolated fungi using morphological and molecular means. I could not identify major flaws in the materials and methods. However, the interpretation of the infection tests and also the presentation of the results in this part should, in my opinion, be revised in a further version of the manuscript. Authors solely interpret their results based on the re-isolation success. However, re-isolation of P. fagi after artificial infection proofed to be difficult and resulted in very low success rates for all infected species. For example F. sylvatica 5 out of 12, F. orientalis 2 out of 30 (? Not so clearly written), Q. petraea 1 out of 10 (?, also here, not clearly written and contradictory to last sentence of the paragraph), C. sativa 0 out of 10. At the same time, they also give the percentage of lesion development for each species. It was high for F. sylvatica and F. orientalis (70 and 74 %, respectivel) and considerably lower for Q. petraea and C. sativa (28 and 19%, respectively). In addition to this, mycopappus formation was only detected on the two Fagus species but not on the others. Their conclusion is that the two Fagus species are clearly succeptible, Q. petraea is a potential host and C. sativa is fully resistant. In my opinion, C. sativa is the least susceptible but nevertheless partially susceptible in vitro because lesions developed around inoculation points in almost 20%. Authors should discuss that Koch’s postulates could not be fulfilled for C. sativa but that it is likely that necroses were caused by P. fagi and that re-isolation failed because such re-isolations proofed to be difficult for all species (and re-isolation of forest pathogens are mostly difficult). They should also elaborate a bit more why re-isolations can be difficult (I was not fully convinced by their endophyte hypothesis). Finally, the results of the infection tests should either be summarized in a table (that would give an easy to understand overview of number of infections, successful infections, number of re-isolations, successful re-isolations and so on) or written more clear with all the necessary numbers. My second point is that the description of the observed infections on C. betulus in nature should be revised and written more clearly. Accoding to their description it was not clear to me whether P. fagi was really the causal agent of these infections. Since no isolations from infected leaves were performed one cannot say much in my opinion. I would revise the sentence in the abstract and only speak about probable damage related to P. fagi or delete the sentence. The same should also be done in the discussion.

29: change to necrotic leaf spots

82: change to molarity should be written in upper case à uM

96: change to “forests”

116: Please also report how many negative controls were prepared per tree species.

Figure 1: It would be nice if you could plot the exact locations of the observed infections with dots. This would give the reader an immediate hint how extensive the “sampling” was.

100-101:Kind of unrealistic: You can’t take the same propagules for molecular/morphological identification and pathogenicity tests. Probably only write that single propagules of infected leaves were identified morphologically and the remaining propagules used for pathogenicity tests.

Fig. 6: Probably also integrate a photo of a lesion. 19% is not cero! I’m surprised about this result. Considering that re-isolation of the fungus from necrotic tissues seems to be difficult, I would not make the conclusion, that C. sativa is totally resistant (see my comments above).

228: Koch’s postulates were also fulfilled for Q. petraea but not for C. sativa or am I wrong?

250: Towards Q. petraea and C. sativa

250-251: change to: New host for P. fagi in Europe.

254-255: write clearer: What did you observe? Did you observe propagules that started lesions? Revise the whole description according to my suggestions above.

257: Change to pathogen (parasite). The description of the observed infections of C. betulus should be revised according to my suggestions above.

265: I’m not sure whether one can say that c. sativa is resistant. Please revise accordingl to my suggestion above.

Author Response

Point-by-point replies to the comments of Reviewer #2

(Legend: Bold = reviewer comment, “>>” = authors reply)

Ogris et al. report the recently detected pathogen P. fagi from Slovenia for the first time. Besides this new first report, authors also performed pathogenicity tests on three different potential host trees of P. fagi and they report, that one of these – Quercus petraea -  is partially susceptible in vitro. In addition, they also report on natural infections of Carpinus betulus near heavily infested Fagus sylvatica trees.

They paper is written and structured quite clear and sound. The English is mostly fine with some minor grammar mistakes here and there. Experiments were carried out according to pathological standards, i.e. they conducted negative controls, made surface sterilizations before isolations and reisolations, identified the isolated fungi using morphological and molecular means. I could not identify major flaws in the materials and methods. However, the interpretation of the infection tests and also the presentation of the results in this part should, in my opinion, be revised in a further version of the manuscript. Authors solely interpret their results based on the re-isolation success. However, re-isolation of P. fagi after artificial infection proofed to be difficult and resulted in very low success rates for all infected species. For example F. sylvatica 5 out of 12, F. orientalis 2 out of 30 (? Not so clearly written), Q. petraea 1 out of 10 (?, also here, not clearly written and contradictory to last sentence of the paragraph), C. sativa 0 out of 10.

>> We wrote clearer the success rate of re-isolations:

sylvatica, Line 180: seven isolates (out of 34) orientalis,Line 203: two pure cultures (out of 30) petraea, Line 217: one pure culture (out of 10) sativa, Lines 233–234: yielded no cultures that would morphologically resemble P. fagi and nine isolates

At the same time, they also give the percentage of lesion development for each species. It was high for F. sylvatica and F. orientalis (70 and 74 %, respectivel) and considerably lower for Q. petraea and C. sativa (28 and 19%, respectively). In addition to this, mycopappus formation was only detected on the two Fagus species but not on the others. Their conclusion is that the two Fagus species are clearly succeptible, Q. petraea is a potential host and C. sativa is fully resistant. In my opinion, C. sativa is the least susceptible but nevertheless partially susceptible in vitro because lesions developed around inoculation points in almost 20%. Authors should discuss that Koch’s postulates could not be fulfilled for C. sativa but that it is likely that necroses were caused by P. fagi and that re-isolation failed because such re-isolations proofed to be difficult for all species (and re-isolation of forest pathogens are mostly difficult).

>> This comment is linked to the comment to the line 265 (see below).

They should also elaborate a bit more why re-isolations can be difficult (I was not fully convinced by their endophyte hypothesis).

>> We rephrased this part in the discussion (lines 253 to 265) and emphasized low success rate of the re-isolations.

Finally, the results of the infection tests should either be summarized in a table (that would give an easy to understand overview of number of infections, successful infections, number of re-isolations, successful re-isolations and so on) or written more clear with all the necessary numbers.

>> We kept the information in the text and we wrote it more clearly with all the necessary numbers.

My second point is that the description of the observed infections on C. betulus in nature should be revised and written more clearly. Accoding to their description it was not clear to me whether P. fagi was really the causal agent of these infections. Since no isolations from infected leaves were performed one cannot say much in my opinion. I would revise the sentence in the abstract and only speak about probable damage related to P. fagi or delete the sentence. The same should also be done in the discussion.

>> We kept the discussion about C. betulus as this is actual observation from the field. We rephrased this part to make it clearer (see comments to the lines 254-255 and 257 below).

29: change to necrotic leaf spots

>> Changed from “leaf nectrotic spots” to “necrotic leaf spots”.

82: change to molarity should be written in upper case à uM

>> Changed to upper case “μM”.

96: change to “forests”

>> Changed from singular to plural “forests”.

116: Please also report how many negative controls were prepared per tree species.

>> Already stated in line 114: “Four non-inoculated healthy leaves of each species represented controls.”

Figure 1: It would be nice if you could plot the exact locations of the observed infections with dots. This would give the reader an immediate hint how extensive the “sampling” was.

>> New Figure 1 was inserted with up-to-date distribution of the disease along with exact locations of observed symptoms in the field. Caption of Figure 1 was changed accordingly to: “Figure 1. Distribution map of Pseudodidymella fagi in Slovenia by regional units of the Slovenia Forest Service along with locations that were checked for presence of the pathogen.”

100-101: Kind of unrealistic: You can’t take the same propagules for molecular/morphological identification and pathogenicity tests. Probably only write that single propagules of infected leaves were identified morphologically and the remaining propagules used for pathogenicity tests.

>> Agreed. We changed the sentence to “For inoculation, first, three to six P. fagi propagules on symptomatic F. sylvatica leaves were morphologically and molecularly identified, and second, the remaining propagules from infected F. sylvatica leaves were carefully transferred…”

Fig. 6: Probably also integrate a photo of a lesion. 19% is not cero! I’m surprised about this result. Considering that re-isolation of the fungus from necrotic tissues seems to be difficult, I would not make the conclusion, that C. sativa is totally resistant (see my comments above).

>> We integrated a photo of a lesion on C. sativa leaf (d). New Fig. 6. was inserted into the manuscript and a caption was added for new photo: “(d) Necrotic leaf spot on the 21st day”. The second part of this comment is linked to the comment to line 265 (see below).

228: Koch’s postulates were also fulfilled for Q. petraea but not for C. sativa or am I wrong?

>> Yes, indeed. We added also Q. petraea: “Therefore, this study is the first confirmation of Koch’s postulates for P. fagi towards F. orientalis and Q. petraea.”

250: Towards Q. petraea and C. sativa

>> We added C. sativa and rephrased the sentence to: “Our in vitro inoculation experiment showed potential pathogenicity towards Q. patraea and C. sativa, which is the first report of new potential hosts for P. fagi in Europe.”

250-251: change to: New host for P. fagi in Europe.

>> Agreed (see comment to the line 250 above).

254-255: write clearer: What did you observe? Did you observe propagules that started lesions? Revise the whole description according to my suggestions above.

>> This section was rephrased as suggested (new line numbers 292–296): “In the field, numerous heavily damaged leaves of Carpinus betulus L. in the immediate vicinity of P. fagi-heavily infected European beech were observed. Macroscopic examination showed that some necrotic lesions on C. betulus leaves developed underneath or in the immediate vicinity of the mycopappus-like propagules of P. fagi.”

257: Change to pathogen (parasite). The description of the observed infections of C. betulus should be revised according to my suggestions above.

>> Changed to pathogen. The description of the observed infections of C. betulus was revised as suggested (see comment to the lines 245–255).

265: I’m not sure whether one can say that c. sativa is resistant. Please revise accordingl to my suggestion above.

>> We agree. We changed lines 289-262 as suggested to: “However, C. sativa seems to be partially susceptible in vitro because 19% of inoculation points developed lesions. Koch’s postulates could not be fulfilled for C. sativa; nevertheless, it is possible that the observed necroses were caused by P. fagi and its re-isolation failed because re-isolations of P. fagi proofed to be difficult for the all tested species.” In Conclusion section line 305 was rephrased: “C. sativa seems to be partially susceptible in vitro.” In Abstract new sentence about this was inserted in lines 17–19: “Although Koch’s postulates could not be fulfilled for C. sativa, it seems to be partially susceptible in vitro because some of inoculation points developed lesions.”